

# Thermal tides in the middle atmosphere at mid-latitudes measured with a ground-based microwave Radiometer

Witali Krochin[1,2], Axel Murk[1,2], and Gunter Stober[1,2]

[1]Institue of Applied Physics, University of Bern
[2] Oeschger Center for Climate Change Research, University of Bern

**Correspondence:** Witali Krochin (witali.krochin@unibe.ch)

**Abstract.** In recent decades, theoretical studies and numerical models of thermal tides have gained attention. It has been recognized that tides have a significant influence on the dynamics of the middle and upper atmosphere, as they grow in amplitude and propagate upwards, they transport energy and momentum from the lower to the upper atmosphere, contributing to the vertical coupling between atmospheric layers. The superposition of tides with other atmospheric waves leads to non-linear wave-wave interactions. However, direct measurements of thermal tides in the middle atmosphere are challenging and often are limited to satellite measurements at the tropics and low latitudes. Due to the orbit geometry such observations provide only a reduced insight into the short-term variability of atmospheric tides. In this manuscript, we present tidal analysis from 5 years of continuous observations of middle atmospheric temperatures. The measurements were performed with the ground-based temperature radiometer TEMPERA, which was developed at the University of Bern in 2013 and was located partially in Bern (46.95°N, 7.45°E) and Payerne (46.82°N, 6.94°E). TEMPERA achieves a temporal resolution of 1-3h and covered the altitude range between 25-55 km. Using an adaptive spectral filter with a vertical regularization (ASF2D) for the tidal analysis, we found maximum amplitudes for the diurnal tide of approximately 2.4 K accompanied by seasonal variability. The maximum amplitude was reached on average at an altitude of 43 km, which also reflected some seasonal characteristics. We demonstrate that TEMPERA is suitable to provide continuous temperature soundings at the stratosphere and lower mesosphere with a sufficient cadence to infer tidal amplitudes and phases for the dominating tidal modes. Furthermore, our measurements exhibit a dominating diurnal tide and smaller amplitudes for the semidiurnal and terdiurnal tides at the stratosphere.

## 1 Introduction

Atmospheric tides are global-scale internal gravity waves forced by solar radiation with periods of an integer fraction of a day (Lindzen and Chapman, 1969; Chapman and Lindzen, 1970; Lindzen, 1979). Thus, tides are classified by the number of oscillations per day as diurnal, semidiurnal, and terdiurnal, respectively. Furthermore, tides can be characterized by their propagation direction and wavenumber. Atmospheric tides that are sun-synchronous are referred to as migrating tides and all other tidal modes are called non-migrating tides. Atmospheric tides are generated by the absorption of solar radiation by water vapor in the troposphere and ozone in the stratosphere (e.g., Sakazaki et al., 2015) and propagate upwards up to the mesosphere/thermosphere. Due to the decreasing density with increasing height, their amplitude grows and they become the



dominating source of variability at the Mesosphere/lower thermosphere (MLT). Atmospheric tides transport energy and momentum in the upper atmospheric layers and enforce layer mixing (Becker, 2017).

During the past decades, there have been many studies on tidal dynamics based on atmospheric modeling (Forbes, 1982; Chang et al., 2008; Hagan et al., 1995, 1999; McCormack et al., 2017; Becker, 2017). Atmospheric tides can be described by normal-mode oscillations in pressure, density, wind, and temperature through Hough-modes (Ortland, 2013). In Sakazaki et al.

(2013) seasonal variations of the migrating diurnal thermal tides are discussed, using Hough-mode decomposition of a global reanalysis data set (MERRA). It was found that the latitudinal-vertical structure is well represented by the four lowest-order Hough modes.

Tidal variations inferred from satellite observations (Zeng et al., 2008; Oberheide et al., 2011b, a; Zhang et al., 2010; Sakazaki et al., 2012; Dhadly et al., 2018), provide global observations but are often not suitable to investigate the short-time variability

due to their orbit geometry. SABER onboard the TIMED spacecraft drifts in local time and, hence, samples every 60 days all local times at each location. Furthermore, SABER has a yaw cycle, which changes the viewing geometry every 60 days and, thus, observes the mid- and polar latitudes on both hemispheres interleaved. Satellites on sun-synchronous orbits, such as MLS onboard AURA (Livesey et al., 2006; Waters et al., 2006), suffer from even more aliasing effects due to fixed local time sampling (Hocke, 2023). Ideally, a temporal resolution of one hour per day is needed to infer the short-time variability of

diurnal or semidiurnal tides. Very often such a high sampling rate is only achieved by ground-based (Baumgarten and Stober, 2019; Krochin et al., 2022a) instruments.

Continuous and high-resolution temperature measurements at the stratosphere and lower mesosphere are sparse. Lidar soundings are often depending on the tropospheric cloud coverage and the daylight capability of the systems. Thus, there are only very limited continuous lidar observations covering several successive days (Baumgarten and Stober, 2019). Furthermore, most

studies of thermal tides focused on the tropical and lower latitude region and the upper mesosphere and thermosphere, where tidal amplitudes are much stronger and become the dominating atmospheric wave (She et al., 2016a; Yuan et al., 2008; She et al., 2016b; Yuan et al., 2021). Only a few observations in the middle atmosphere (stratosphere and mesosphere) are documented in the literature (Gille et al., 1991; Kopp et al., 2015; Fong et al., 2022). However, most meteorological reanalysis update the data assimilation every 6 hours, although the model output is provided at a higher cadence (Gelaro et al., 2017). Due

to the sparsity of observations at the stratosphere, some observations are only available every 12 hours (e.g., radiosondes), and, thus, measurements that capture the short-term tidal variability at these altitudes are crucial to constrain the tidal amplitudes and phases and also to infer heating rates due to the absorption of solar radiation by ozone and water vapor.

In this manuscript, we present diurnal, semidiurnal, and terdiurnal thermal tide amplitudes at altitudes between 20-55 km, derived from continuous long-term observations of atmospheric microwave spectra by a ground-based microwave radiometer

(TEMPERA, Stähli et al. (2013)) located at the Meteo Swiss technical center in Payerne (46°48.0′ N, 6°56.0′ E;) and discuss the seasonal and latitudinal climatology. In sections 2,3 and 4 the instrument, the retrieval method for temperature profiles, and the resulting temperature time series are described in more detail. In sections 5 and 6 the tidal analysis method and the resulting tidal amplitudes are presented and in section 7 the results are discussed.



## 2 Instrument description

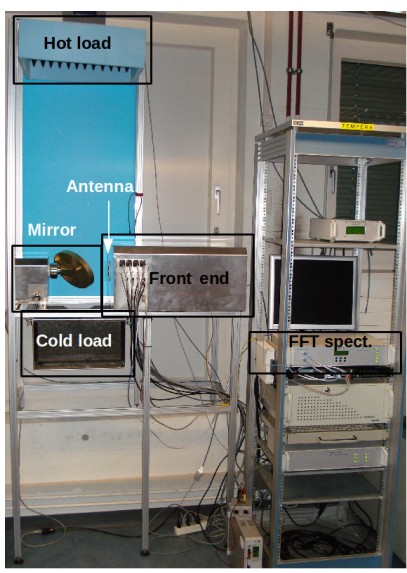

**Figure 1.** TEMPERA at the Institute of Applied Physics at the University of Bern. Navas-Guzmán et al. (2015)

The TEMperature RAdiometer (TEMPERA) was built at the University of Bern in 2013 (Stähli et al., 2013). TEMPERA measures microwave emission from atmospheric oxygen at the 60 GHz oxygen emission complex (see Figure 2). With a spectral resolution of 30 kHz and a bandwidth of $2\times$ 480 MHz, the fine structure of rotational transitions, used to retrieve temperature in the middle atmosphere, can be resolved (see Figure 3). The original operational mode uses 12 additional filter banks and scans at different zenith angles to retrieve tropospheric temperatures. In this operational mode, the temporal resolution for

stratospheric spectra is 3 hours. TEMPERA operated continuously in this measurement mode during the years 2013-2018 at the Meteo Swiss Technical Center in Payerne. Stratospheric spectra are calibrated with a two-point calibration, where an internal noise diode and an ambient load are used. The noise diode is calibrated once a month with liquid nitrogen (LN2) (see Figure 1. The fine structure spectra are inverted using ARTS (Atmospheric Radiative Transfer Simulator, Eriksson et al. (2005)). A more detailed technical description is found in Stähli et al. (2013); Krochin et al. (2022a).




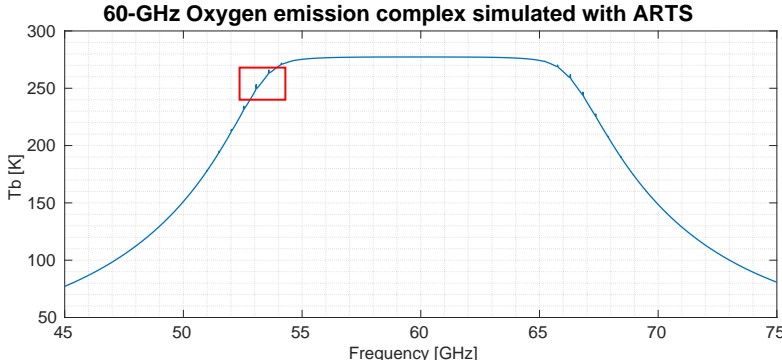

**Figure 2.** The Oxygen emission complex, simulated with ARTS for the location of Bern during Winter at a zenith angle of 30°. The measurement range of TEMPERA is illustrated with a red rectangle.

In 2022 an updated retrieval algorithm was published (Krochin et al., 2022a), which accounted for the Zeeman-splitting in the line center due to the coupling of atmospheric oxygen to the Earth's magnetic field. The update improved the altitude resolution and increased the upper altitude retrieval limit defined by the measurement response. The basis for this improved retrieval was provided in the ARTS software, which included a module for the Zeeman-Splitting. In the same year, TEMPERA

was relocated to the Institute of Applied Physics at the University of Bern. This change was accompanied by a new measurement mode dedicated to stratospheric and lower mesospheric soundings. The new mode relies on noise diode calibration and avoids spending measurement time to perform a tipping curve for tropospheric retrievals, maximizing the measurement time of stratospheric spectra resulting in a temporal resolution of about 1 hour. In addition, several minor updates of the retrieval algorithm were performed, including an improved tropospheric correction, apriori error matrix, baseline correction, retrieval

for frequency shift and frequency stretch, and filtering of contaminated spectra (see section 3).





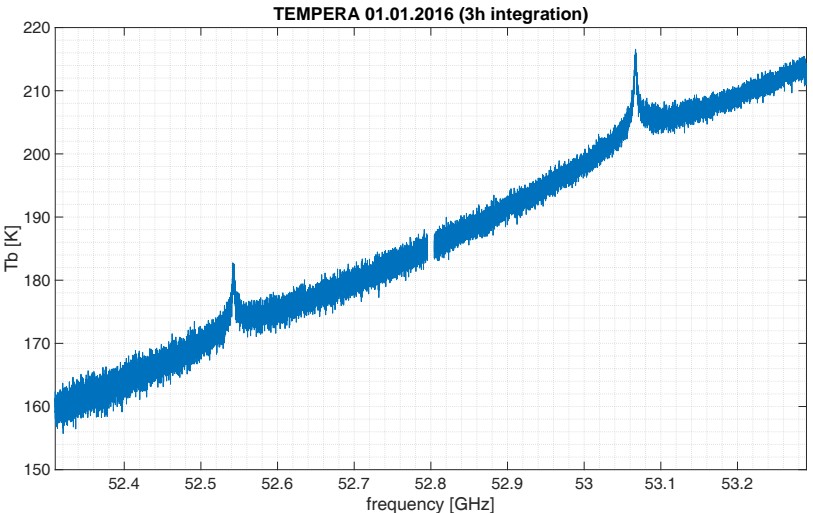

**Figure 3.** Spectrum of atmospheric Oxygen fine structure emission lines measured with TEMPERA after 3h of integration time (00.00 - 03.00). The ambient load in combination with the integrated noise diode, was used for calibration of this spectrum.

## 3 Temperature retrievals from atmospheric spectra

The first step is to set up a model atmosphere and simulate the forward radiative transfer.

$$\mathbf{y} = \mathbf{F}(\mathbf{x}) + \epsilon \tag{1}$$

Where $\mathbf{y}$ is the measurement vector, $\mathbf{F}$ the forward model, $\mathbf{x}$ the atmospheric state vector, and $\epsilon$ the measurement error. Following the formalism of Rodgers (2000) the forward model is inverted by an optimal estimation method. Assuming that $\mathbf{y}$ and $\mathbf{x}$ have Gaussian probability distributions $P(\mathbf{x})$ and $P(\mathbf{y})$, and using the Bayes' theorem

$$P(\mathbf{x}|\mathbf{y}) = \frac{P(\mathbf{y}|\mathbf{x})P(\mathbf{x})}{P(\mathbf{y})}, \tag{2}$$

a cost function of the following form can be found

$$J(\mathbf{x}) = -2\ln P(\mathbf{x}|\mathbf{y}) = [\mathbf{y} - \mathbf{F}(\mathbf{x})]^T \mathbf{S}_\epsilon^{-1} [\mathbf{y} - \mathbf{F}(\mathbf{x})] + [\mathbf{x} - \mathbf{x}_a]^T \mathbf{S}_a [\mathbf{x} - \mathbf{x}_a] \tag{3}$$

where $J(\mathbf{x})$ is minimized by a Levenberg-Marquardt algorithm. Here $\mathbf{x}_a$ is the apriori state, used as the initial guess to start the iteration, and $\mathbf{S}_a$ and $\mathbf{S}_\epsilon$ are the apriori and measurement covariance matrices. An important quantity for the retrieval analysis is the averaging kernel

$$\mathbf{A} = \mathbf{GK} \tag{4}$$

and the corresponding measurement response

$$\mathbf{MR} = \mathbf{AI}, \tag{5}$$





where $\mathbf{K}$ is the weighting function matrix

$$\mathbf{K} = \frac{\partial \mathbf{F}(\mathbf{x})}{\partial \mathbf{x}}, \tag{6}$$

and $\mathbf{G}$ denotes the gain matrix

$\quad \mathbf{G} = \left[\mathbf{K}^T \mathbf{S}_\epsilon^{-1} \mathbf{K} + \mathbf{S}_a^{-1}\right]^{-1} \mathbf{K}^T \mathbf{S}_\epsilon^{-1}, \tag{7}$

and $\mathbf{I}$ denotes the unit matrix. The optimal estimation technique contains several sources of uncertainty, which are the observation error $\mathbf{S}_O$, the smoothing error $\mathbf{S}_S$, which both contribute to the total error $S_{tot}$

$$\mathbf{S}_O = \mathbf{G}\mathbf{S}_\epsilon \mathbf{G}^T \tag{8}$$

$$\mathbf{S}_S = [\mathbf{A} - \mathbf{I}]\,\mathbf{S}_a\,[\mathbf{A} - \mathbf{I}]^T \tag{9}$$

$\quad \mathbf{S}_{tot} = \mathbf{S}_O + \mathbf{S}_S \ . \tag{10}$

Within ARTS the Zeeman Splitting module (Larsson et al., 2019) was used to account for the Zeeman effect in the forward model. A detailed description of the forward model retrieval algorithm is found in Krochin et al. (2022a). We implemented only a minor change related to the tropospheric correction. Instead of using the method suggested in Ingold and Kämpfer (1998), the tropospheric opacity is retrieved in ARTS using the spectral data of the line wings. Minor improvements were reached by

including, frequency shift and frequency stretch in the ensemble of retrieved quantities, and by decreasing the apriori error of the retrieved baseline. The temperature profile apriori error was set constant to a brightness temperature covariance of 15K. Figure 4 illustrates the measurement response (MR), the averaging kernel (AVK), total retrieval error, and the full-width-at-half-maximum of the rows of the AVK matrix, conventionally referred to as altitude resolution. We consider only altitudes with a measurement response of MR>0.6 for all further scientific analyses. A lower measurement response indicates that the

solution at these heights is more and more dominated by the apriori state. The forward model formalism, the retrieval algorithm, and the definitions of the retrieval quantities following the formalism from Rodgers (2000).

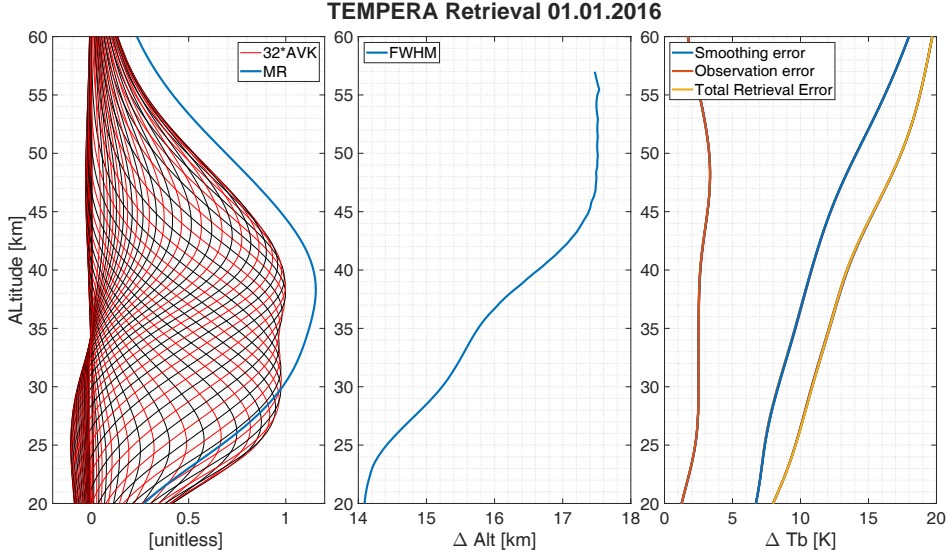

**Figure 4.** The left panel shows the AVK matrix for each altitude. The AVK values were inflated by 32 to get them to a comparable scale to the MR in the same plot (ARTS output). The middle panel visualizes the FWHM of the AVK matrix for each altitude, which corresponds to the altitude resolution. The right panel exhibits all three retrieval errors as explained in Eq. 8 to 10.

## 4 TEMPERA observations

In this manuscript, we present measurements of altitude and time-resolved profiles of atmospheric temperatures, retrieved from continuous TEMPERA observations conducted between 2014 to 2023. Two periods are to be distinguished due to a change
in the operational mode made in 2022. Between 2014-2017 TEMPERA operated in an interleaved observation mode sharing the measurement time between a tropospheric mode, which scanned several zenith angles to retrieve tropospheric temperatures (Stähli et al., 2013), and a stratospheric/lower mesospheric mode measuring atmospheric spectra at a zenith angle of 30°, which resulted in an effective temporal resolutions of 3 hours. The operational mode implemented in 2022 employs only one zenith angle at 30°, which improves the time resolution of stratospheric spectra to 1 hour. Figure 3 illustrates a calibrated spectrum,
observed at the Meteo Swiss Technical Center Payerne in 2016.

The time series of retrieved temperature profiles is shown in Figure 5. The blank times are mainly caused by liquid water clouds or rain that attenuate the stratospheric signal. Under clear sky conditions, the observed frequency range is less affected by the emission/absorption of $H_2O$ molecules and only the effects of line mixing remain. However, liquid water clouds create thermal emission, which increases the brightness temperatures over the entire frequency band at our receiver. An approach
to mitigate this problem is to include a cloud box in the forward model of the retrieval algorithm. This method improves tropospheric temperature retrievals (Bernet et al., 2017), but was not tested on stratospheric retrievals and, thus, is not included in the current version of the stratospheric retrieval.





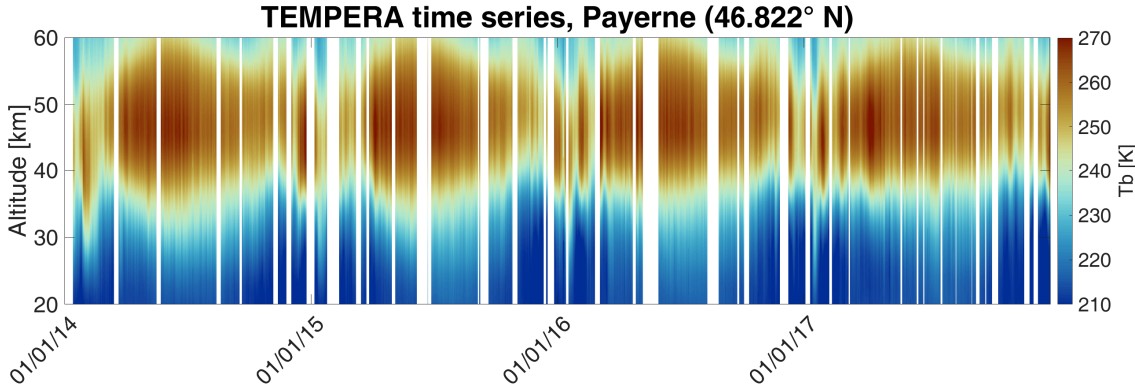

**Figure 5.** Continuous series of temperature profiles, retrieved from TEMPERA measurements. Data gaps are mainly due to weather conditions. The altitude range is 25-55 km (MR>0.6) but the range in the plot is 20-60 km for illustration.

## 5 Tidal Analysis - Adaptive Spectral Filter (ASF)

Previous studies have shown that the prevailing tidal modes at mid-latitudes are migrating tides (Stober et al., 2020b; Baum-
garten and Stober, 2019; Hibbins et al., 2019). Non-migrating tides have often smaller amplitudes compared to their migrating counterpart. However, satellite observations have demonstrated their presence at the lower latitudes (Oberheide et al., 2011a). TEMPERA observations are only available at one geographic location and, thus, in all further analysis, we refer to the tides as total tides consisting of the migrating and non-migrating modes. Furthermore, we classify the different tidal modes only by their period and discuss diurnal, semidiurnal, and terdiurnal total tidal amplitudes and phases. Local tidal oscillations are often
modeled by a mean state and a simple superposition of sinusoidal functions for each included period;

$$T(t_k) = T_{0k} + \sum_{n=1}^{3} \left[ a_{nk} \sin\left(\frac{2\pi}{P_n} t_k\right) + b_{nk} \cos\left(\frac{2\pi}{P_n} t_k\right) \right]. \tag{11}$$

Where $T_{0k}$ is the background state (median over $k$-th window), $P_n = [1, 1/2, 1/3] \times$ day, the period, $A_{nk} = \sqrt{a_{nk}^2 + b_{nk}^2}$ the amplitude, and $t_k$ is the local time of the $k$-th window. The phase shift $\Delta\phi_{nk}$ (hours from midnight) can be derived by

$$\Delta\phi_{nk} = \arctan\left(\frac{b_{nk}}{a_{nk}}\right) \tag{12}$$

In this study, we applied the adaptive spectral filter (ASF) technique that was already used in many studies and on many different data such as lidar observations, meteor radar winds, EISCAT ion velocity, or GCM winds and temperatures (Stober et al., 2017; Baumgarten and Stober, 2019; Stober et al., 2020b, 2021; Günzkofer et al., 2022).

The ASF also includes a vertical regularization to ensure a smooth phase behavior of each tidal mode, which seems to reduce the contamination due to gravity waves with shorter vertical wavelengths. Furthermore, the algorithm adapts the window length





for each fitted tidal mode to capture transient events that alter the tidal amplitude and phase. Another benefit of this technique is that data gaps or unevenly sampled time series can be analyzed as well. More details can be found in Baumgarten and Stober (2019); Stober et al. (2020b). Here, we implemented the ASF using a 4-day sliding window after the removal of the median. The error propagation for the tidal amplitudes and phases was calculated by weighting the least square error, with the total retrieval error.

## 6 Results of tidal analysis


Applying ASF on the data set of continuous temperature profiles results in 3 sets of continuous-time and altitude-resolved amplitude and phases (Local Solar Time of maximum). For the period 2014-2017, the time resolution (3h) is not sufficient to resolve semi-diurnal and terdiurnal tides, therefore only one single set of diurnal tides was analyzed. For the period 2022-2023 however, the time resolution (1 h ) is sufficient to resolve semi- and terdiurnal tides.






**Figure 6.** a) Vertical resolved temperature measurements observed with TEMPERA for the period from 24.03.2014-07.04.2014. b) Inferred absolute Temperature anomalies retrieved using a 4-day sliding median to visualize short-scale variations. c) Vertical integrated temperature anomalies calculated from measurements between 40-55 km and a corresponding best-sine-fit. d) Retrieved diurnal tide amplitude profiles calculated from ASF.

Figure 6 shows temperature profiles (top panel) and the corresponding tidal amplitudes (bottom panel) for 14 days during spring 2014. The temperature anomaly (second panel from the top) is the difference between the raw temperature and a background state, smoothed with a 4-day sliding window (median). It shows the complex dynamic at the stratosphere with





alternating warmer and colder periods at different altitudes. The third plot from the top shows a median over the anomaly
profile from 40-55 km. For illustration purposes, a sine function with a period of 24 hours was fitted to the curve. Deviations
from the sine fit are assumed to be higher period oscillations caused by gravity waves and planetary waves. Also, the period
of thermal tides is not stable and can slightly change from day to day (Baumgarten and Stober, 2019; Stober et al., 2020b; van
Caspel et al., 2023).

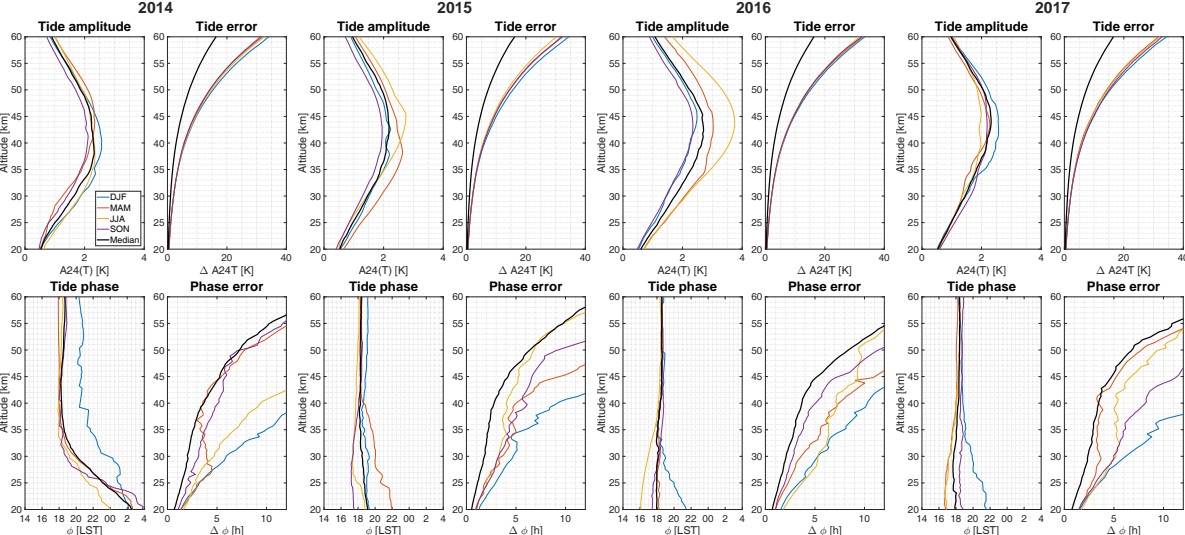

**Figure 7.** Top: Seasonal averaged diurnal tide profiles with the corresponding error. The black curve represents the median over
a whole year. Bottom: Seasonal averaged diurnal tide phase profiles with the corresponding error.

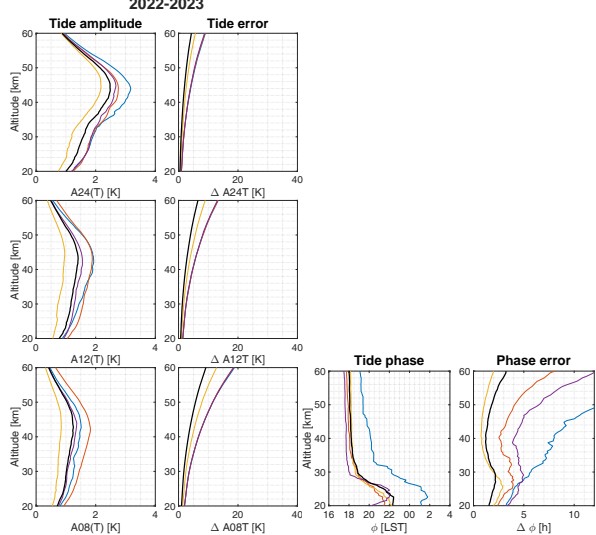

**Figure 8.** Left: Seasonal averaged diurnal, semi diurnal, and terdiurnal tide profiles with the corresponding error (as in Figure
7). Right: Seasonal averaged diurnal tide phase profiles with the corresponding error (see section 5).



| season | 2014 Max A24T(@ z [km]) | 2015 Max A24T(@ z [km]) | 2016 Max A24T(@ z [km]) | 2017 Max A24T(@ z [km]) |
|---|---|---|---|---|
| DJF | 2.6(40) | 2.2(43) | 2.5(45) | 2.6(42) |
| MAM | 2.3(44) | 2.6(38) | 3.0(43) | 2.3(43) |
| JJA | 2.5(38) | 2.8(45) | 3.7(44) | 2.0(40) |
| SON | 2.1(41) | 2.0(42) | 2.3(43) | 2.2(43) |
| full median | 2.4(38) | 2.4(43) | 2.8(42) | 2.3(44) |

**Table 1.** Maxima and corresponding altitudes of seasonal averaged diurnal tidal amplitudes between 2014-2017.

| season | diurnal Max A24T(@ z [km]) | semidiurnal Max A12T(@ z [km]) | terdiurnal Max A8T(@ z [km]) |
|---|---|---|---|
| DJF | 3.1(45) | 1.9(44) | 1.5(43) |
| MAM | 2.8(44) | 1.9(43) | 1.7(42) |
| JJA | 2.2(45) | 1.0(45) | 0.8(43) |
| SON | 2.6(44) | 1.4(41) | 1.2(44) |
| full median | 2.4(44) | 1.4(43) | 1.2(44) |

**Table 2.** Maxima and corresponding altitudes of seasonal averaged diurnal, semidiurnal, and terdiurnal tidal amplitudes.

Yearly and seasonal averaged amplitude profiles for the period 2014-2017 are illustrated in Figure 7 (upper panels). The black line shows the median for an entire year and colored lines represent medians over the corresponding seasons (December-January-February (DJF) (blue), March-April-May (MAM)(red), June-July-August (JJA)(orange), and September-October-November (SON)(purple)). The mean tide amplitude profile has a maximum between 2.0-3.7 K at 38-45 km (see also Table 1). The seasonal averages show an increased tidal activity in the spring months for the years 2015, and 2016, whereas in 2014 and 2017 the highest tidal activity was found during winter. The absolute difference between summer and winter tide amplitudes is, however, much smaller than the corresponding standard deviation and, thus, is insignificant. The lower panels show the tide phases, which in this case are the local solar times of the tide peak. From 20-30 km, a downward phase propagation can be seen in 2014 with an estimated phase speed of -1.6 km/h. At 30-35 km the phase propagation turns to a constant value of 18 LST. The reversal of the phase propagation seems to be related to the ozone diurnal cycle and will be discussed later. Figure 8 (left panel) shows averaged tidal amplitudes for the period 2022-2023. The maximum for diurnal tide amplitudes (upper panel) occurs between 41-45km where the amplitude reaches values between 2.2-3.2 K. Due to the instrument updates between 2017 and 2022, we analyzed also the semidiurnal (middle panel) and terdiurnal (lower panel) tidal amplitudes for these observations. The profiles show maximum values of 1-1.9 K around 41-45 km for semidiurnal and 0.8-1.5 around 42-44km for terdiurnal tide (see Table 2). Diurnal and semidiurnal tides have the highest amplitudes during winter time, while the terdiurnal tide exhibits maximum amplitudes in spring. The phases for the diurnal tides are shown in Figure 8 (right panel). Semidiurnal and terdiurnal

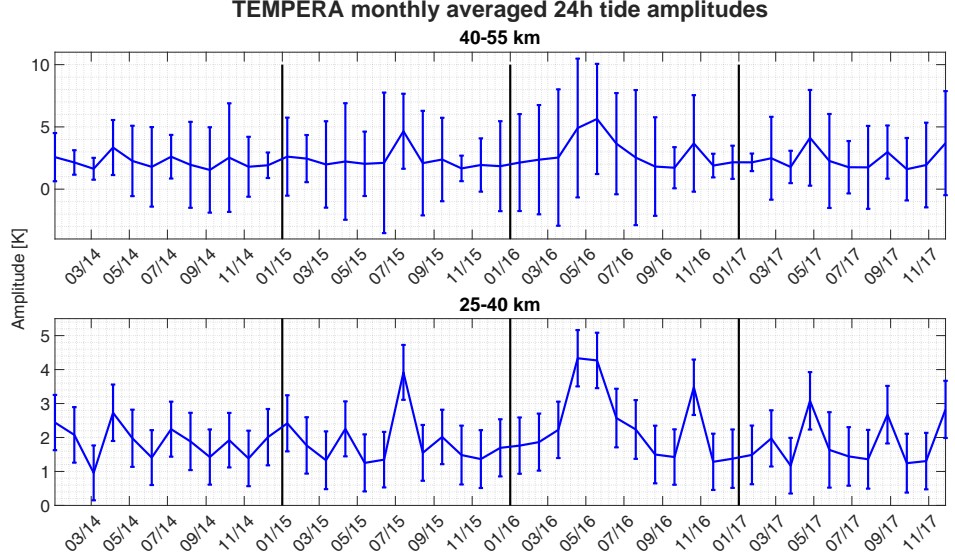

**Figure 9.** Monthly (31-day window) and altitude (15 km) averaged diurnal tide series with corresponding atmospheric variability (see section 7).

phases are not shown. These appeared to be very variable, which is partly attributed to the much smaller amplitudes and also due to other geophysical effects, which mask the tidal signature such as gravity waves. The phase propagation of diurnal tides

shows a reversal from upward to downward at 22 km, a downward propagating phase with a phase speed of -1.8 km/h between 22-28 km, and a constant phase above 30 km.

To illustrate the seasonal climatology, the tide amplitudes were averaged over two altitude regions. The lower region is between 25-40 km and the upper region is between 40-55 km. Figure 9 shows the time series of the period 2014-2017 for both

altitude regions. A significantly increased tidal activity appeared during the spring for the years 2016, 2017, and summer 2015, whereas for the year 2014 no significant seasonal pattern was found. The years 2016 and 2017 exhibit also a second maximum in autumn. Figure 10 shows the same plot for the period 2022-2023. All tidal amplitudes show increased activity in September, December, and April.

## 7 Discussion

Continuous temperature measurements at the stratosphere and lower mesosphere are crucial for benchmarking of reanalysis models or meteorological analysis. Such observations are beneficial for cross-comparison and also for data assimilation. Understanding the short-term tidal variability and vertical propagation requires higher cadence observations and more data assimilation (Liu, 2016; McCormack et al., 2017; Stober et al., 2020b; Dhadly et al., 2018; van Caspel et al., 2023). Currently, MERRA2 meteorological analysis is provided with hourly or three-hourly temporal resolution. However, the 3DVAR data as-



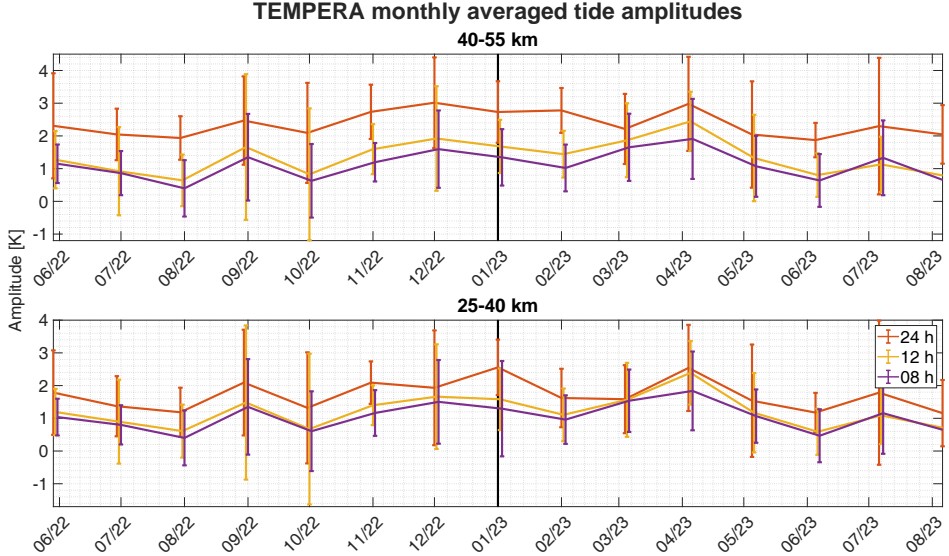

**Figure 10.** Monthly (31 day window) and altitude (15 km) averaged diurnal, semi diurnal, and terdiurnal tide series.

simiation is performed every 6 hours (Gelaro et al., 2017). Furthermore, some measurements that are assimilated in MERRA2 at the stratosphere and the lower mesosphere are only available every 12 hours (e.g., radiosondes). The sparsity of measurements at the stratosphere and lower mesosphere will further increase due to the end-of-life-time of some satellite instruments such as MLS (Waters et al., 2006; Livesey et al., 2006).

Previous studies of atmospheric temperature tides at the stratosphere and lower mesosphere at mid-latitudes were based on
lidar observations having full daylight capabilities. However, lidar observations depend on tropospheric cloud coverage, and, thus, the obtained lidar monthly climatologies were inferred by constructing multiyear composites of a day. Measurements acquired from different years and of various lengths covering full days or only a few hours were stacked together and analyzed to estimate tidal amplitudes from the obtained mean composite day (Kopp et al., 2015). So far, there was only one study that covered a 10-day continuous lidar measurement, which showed some tidal intermittency at the stratosphere (Baumgarten et al.,
2018; Baumgarten and Stober, 2019). Hence, the shown TEMPERA measurements of stratospheric and lower mesospheric temperatures are valuable to investigate and continuously monitor the source variability of tides.

Diurnal tide profiles reach their seasonal maxima at an altitude around 38-45 km with values around 2-3.7 K, semidiurnal tides at an altitude around 42-45 km with maximum amplitudes of 1-1.9 K, and terdiurnal tides become maximal at 42-43 km with an amplitude of 0.8-1.5 K. This corresponds well to the results from Kopp et al. (2015), where the measurements were performed
with a Lidar located in Kühlungsborn (Germany, 54.1469° N, 11.7420° E). However, the seasonal pattern in Kopp et al. (2015) shows a maximum of the diurnal tide at 45-55 km altitude in March and October with significantly lower amplitudes during summer. The results in our study show partially a similar behavior, but not for all years. We found the largest tidal amplitudes in May and October 2016, May and September 2017, September 2022, and April 2023. This follows roughly the pattern of maximal tidal activity in Spring and Autumn, where the spring maximum appears between the end of April and the end of May,





and the autumn maximum appears between the beginning of September and the end of October. However, the tidal activity in the years 2014 and 2015 deviates from this climatological pattern. We attribute these differences in the first years 2014-2017 of TEMPERA measurements to instrumental effects and many changes in the measurement mode. Since 2022 TEMPERA has operated in a dedicated stratospheric and lower mesospheric measurement mode and improved retrievals are implemented (Krochin et al., 2022a), which solved most of the potential problems of the first light observations during the development

phase of the prototype.

Classical tidal theory predicts an increased diurnal tidal amplitude during the summer months due to the increased solar heating (Lindzen and Chapman, 1969; Lindzen, 1979). However, this was only confirmed for the year 2015. Figure 9 and 10 show the standard deviation of the corresponding tide amplitudes in the error bars. The standard deviation of the tidal amplitudes is a measure of their geophysical variability and intermittency. Our observations were conducted in Bern and Payerne at the Swiss

Plateau, which is next to the Alpine main ridge, a region that is often affected by strong summer convective instabilities due to thunderstorms. These convective cells excite gravity waves, which propagate into the stratosphere and alter the circulation by depositing their energy and momentum likely triggering multistep vertical coupling processes (Becker and Vadas, 2018; Vadas and Becker, 2018; Vadas et al., 2023). Although the averaged tide amplitudes exhibit only partially the expected pattern for the entire period, the geophysical variability indicates good agreement with the tidal theory. This variability reaches a minimum

around winter time (March 2014, December 2015, October 2016, February 2017, February 2023). Also, it appears that there are two periods per year (around Mai and October) where it is likely to find increased tidal variation. In general, the variation in summer is 2-3 K higher than the tidal variability found during the winter months.

Observations of tidal phases at the stratosphere are rare. Due to the small tidal amplitudes phases appear to be very noisy and variable. This is also the case for TEMPERA measurements. The vertical profiles of the phases exhibit two regions. Above an

altitude of 35 km the phase turns to a constant value of about 18 LST. Below 35 km the phases are much more variable and sometimes tend to reflect clear signs of a phase progression. A constant phase with altitude suggests that the tides are forced at these altitudes, whereas a changing phase indicates tidal propagation from below.

The increased phase variability and inversion of the phase progression at 35 km appear to be coupled to the ozone diurnal cycle. At this altitude, the relationship between ozone and temperature changes from a positive to a negative correlation due

to a change in the chemical balance. We illustrate this effect by calculating a correlation between the ozone volume mixing ratio (VMR) and our temperature tidal amplitudes. Leveraging the ozone VMR measurements retrieved from the GROMOS radiometer (Sauvageat et al., 2022), which is located next to TEMPERA, we investigate their altitude-dependent relationship. Figure 11 shows Pearson Correlation coefficients, estimated over a 30-day sliding window, after re-gridding on a generic altitude grid. Since heating due to the absorption of solar radiation by ozone is the main tidal forcing mechanism at these altitudes,

changes in ozone VMR are expected to be reflected in our tidal amplitude measurements. A similar calculation as in Figure 11 was also shown in (Sauvageat et al., 2023). A more detailed analysis of the ozone diurnal cycle was already performed in Schranz et al. (2018) and provides a pathway to future analysis.



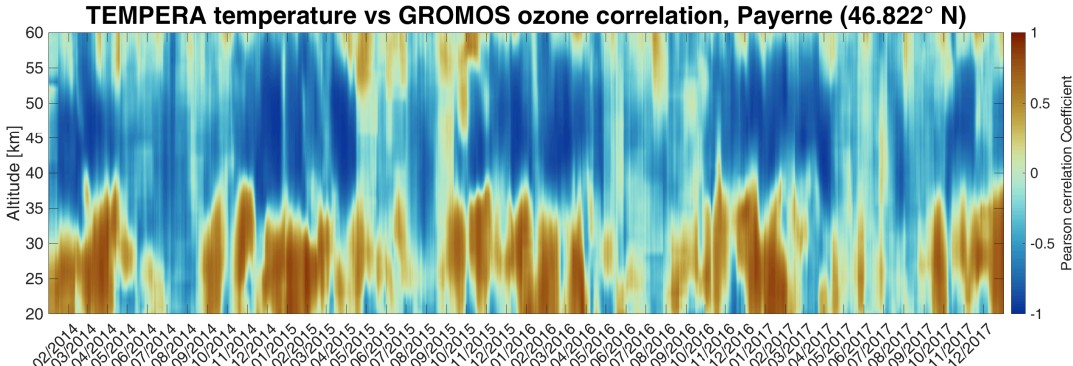

**Figure 11.** Pearson Correlation coefficients between TEMPERA temperatures and GROMOS Ozone measurements. Correlation Coefficients were calculated over a 30-day sliding window after interpolation on a generic altitude grid.

The main challenges for this type of analysis are related to instrument noise and data gaps, which complicate the spectral
analysis and influence the phase information. In addition, there is some tidal intermittency due to planetary wave activity or strong gravity wave excitations (e.g., thunderstorms), which can lead to data gaps rendering classical Fourier transform or Wavelet techniques for the spectral analysis not applicable, or even useless. The ASF2D implemented for the TEMPERA tidal analysis, is designed to provide a decomposition of time series with data gaps or unevenly sampled measurements including error propagation and even a vertical regularization (see Stober et al. (2020a) and Baumgarten and Stober (2019)). Therefore,
this technique seems to be adequate to be applied for radiometric high temporal resolution tidal studies.

## 8 Conclusions

We demonstrated the new capabilities of TEMPERA, the University Bern temperature radiometer, to perform continuous temperature soundings at the stratosphere and lower mesosphere. A recently implemented observational mode dedicated to stratospheric and mesospheric measurements together with updated retrievals (Krochin et al., 2022a) permit temperature mea-
surements between 20-60 km altitude with a temporal resolution of 1 hour. The instrument operates autonomously and requires only occasional LN2 calibrations. The radiometer performs nominally also under tropospheric cloud cover conditions and only precipitation or thick liquid water clouds can lead to a loss of the stratospheric signal from both spectral lines.

Based on these continuous temperature soundings, we derived diurnal, semidiurnal, and terdiurnal tidal amplitude profiles between 20-55 km between 2014-2023. Amplitude profiles reach the maximum at an altitude range from 40-45 km with am-
plitudes of 2-3.7 K. The obtained tidal amplitudes agree with previous results from multiyear composite lidar observations at mid-latitudes Kopp et al. (2015). Our continuous temperature soundings indicate a notable year-to-year variability of the seasonal tidal activity, which only partially reflects the predictions from the classical tide theory (maximum amplitudes during summer). However, the amplitude geophysical variability appears to be increased by 2-3 K during the summer compared to the winter season, which is likely caused by summer tropospheric convection above the Swiss Plateau and the corresponding



multistep vertical coupling due to gravity waves inducing changes in the stratospheric circulation due to the deposition of the transported energy and momentum.

The retrieved tidal phase also exhibits a characteristic vertical structure. Above 35 km, diurnal tidal phases appear to be constant suggesting a direct excitation of the tide due to the ozone absorption. Below 35 km, we observed an increased variability and in some years a vertical phase progression indicating a vertical propagation of a tide that was forced below. This is confirmed

by the Pearson correlation of ozone-temperature measurements, which indicates a change from positive to negative correlation at this altitude.

Another big advantage of TEMPERA is the instrument cost compared to lidars with full daylight capability. The lower unit costs provide an opportunity to install larger observational networks. A larger network of ground-based radiometers would greatly improve the tidal analysis. By deploying TEMPERAs at four different measurement locations, wavefront direction,

orientation, and propagation velocity could be resolved. Such an observational network could be complemented with wind observations from the wind radiometers WIRA (Hagen et al., 2018, 2020). Furthermore, the Microwave Group of the University of Bern is currently developing a new fully polarimetric temperature radiometer (TEMPERA-C, Krochin et al. (2022b)). The new instrument will have an increased altitude range (up to 60km) and an even better time resolution of about 30 min.

*Data availability.* MERRA-2 data are available at MDISC, managed by the NASA Goddard Earth Sciences (GES) Data and Information

Services Center (DISC) DOI:10.5067/QBZ6MG944HW0. TEMPERA temperatures are shared on request (gunter.stober@unibe.ch).





## Appendix A: MERRA2 Comparison

For the location of Payerne (46°49'N 6°56'E) MERRA2 temperature profiles for the period 2014-2017 were analyzed with the method described in section 5. The results are illustrated in Figure A1. The amplitude profiles have maximal values of 2-3.4 K at altitudes 47-53 km. Diurnal tide amplitudes and variations also show maximal activity during winter months (Figures A1

and A2).

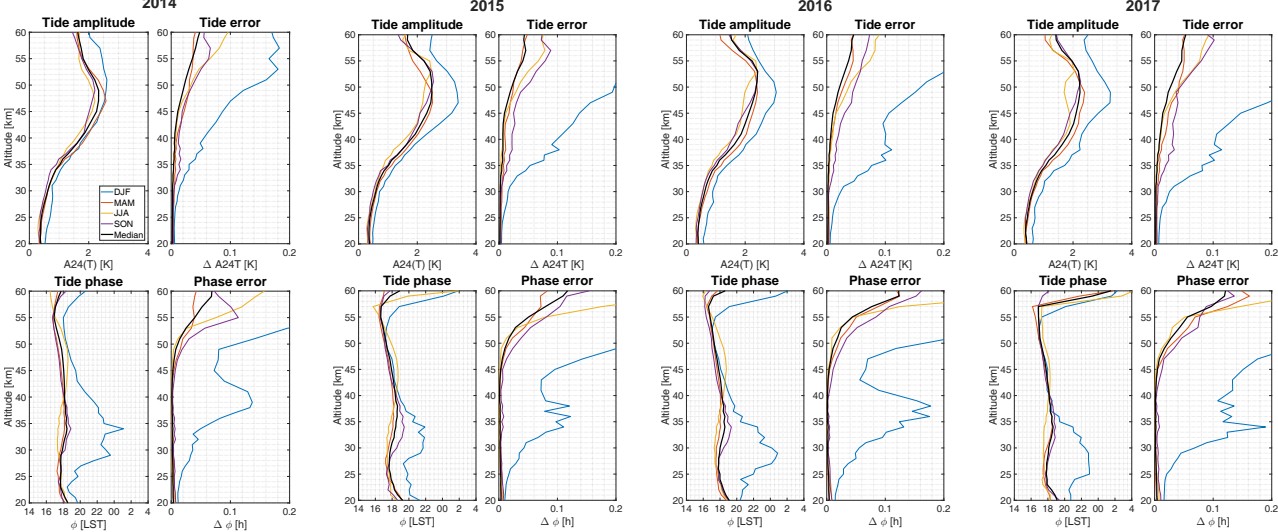

**Figure A1.**

Top: Seasonal averaged diurnal tide profiles with the corresponding error (same as Figure. 7) derived from MERRA2 data. The black curve represents the median over a whole year.

Bottom: Seasonal averaged diurnal tide phase profiles with the corresponding error.

While MERRA2 tide profile maxima match well with our results, the altitude where the maxima are reached is roughly 6 km higher as in our case. Phase profiles show anomalous behaviour between 25-35 km mostly in winter months. Downward phase propagation was found in all years, but only up to 25km. Above 35 km the phase is not constant but rather shows downward

propagation and turns sharply upward above 55 km. However, up to 55 km the phase remains roughly around 18 LST, except in winter months.

The seasonal pattern of MERRA2 diurnal tide amplitudes deviates from our results and classical tide theory. The highest amplitude and variation appear from December to February, while the lowest activity was found in summer. We assume that

the main reason for this discrepancy is the implementation of the ozone diurnal cycle in the reanalysis. However, MERRA2 is a meteorological reanalysis and thus the quality of the obtained wind and temperature fields depends crucially on the data to be assimilated. Above 35 km altitude, the data coverage becomes more sparse, and the temporal resolution of the assimilated data products is no longer sufficient to constrain tidal amplitudes and phases on a global scale.





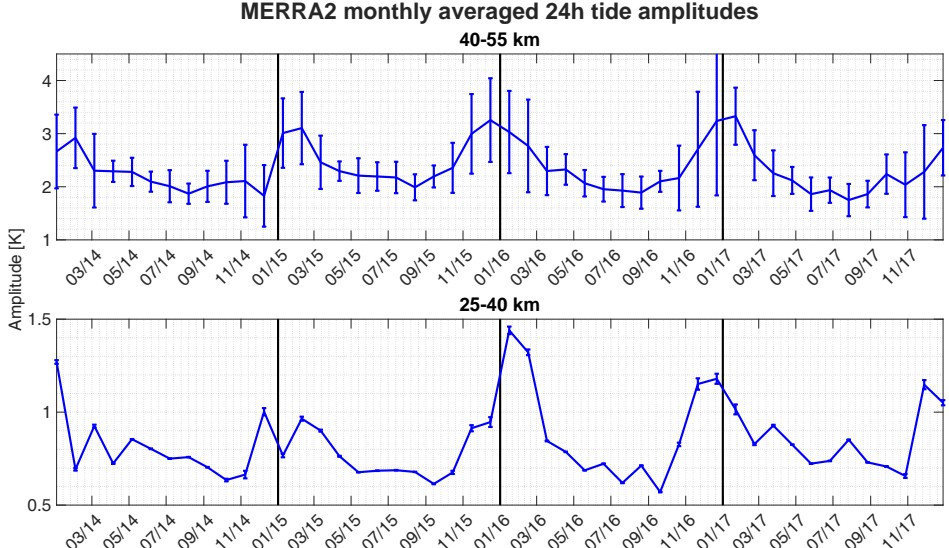

**Figure A2.** Monthly (31 day window) and altitude (15 km) averaged diurnal tide series (same as Figure. 9), derived from MERRA2 data.

| season | 2014 | 2015 | 2016 | 2017 |
| --- | --- | --- | --- | --- |
| | Max A24T(@ z [km]) | Max A24T(@ z [km]) | Max A24T(@ z [km]) | Max A24T(@ z [km]) |
| DJF | 2.6(51) | 3.4(47) | 3.1(49) | 3.3(49) |
| MAM | 2.6(47) | 2.5(47) | 2.4(49) | 2.4(49) |
| JJA | 2.2(47) | 2.5(53) | 2.4(53) | 2.0(53) |
| SON | 2.2(49) | 2.5(51) | 2.4(53) | 2.3(51) |
| full median | 2.4(49) | 2.7(49) | 2.6(53) | 2.5(49) |

**Table A1.** Maxima and corresponding altitudes of seasonal averaged diurnal tide profiles derived from MERRA2 data.

*Author contributions.* WK and GS conceptualized the content of the manuscript. WK implemented the retrieval and performed the data anal-

ysis of TEMPERA observations. GS reduced the MERRA2 data for the validation. All authors contributed to the editing of the manuscript.

*Competing interests.* The authors declare that they have no competing interests.

*Acknowledgements.* This research has been supported by the Schweizerischer Nationalfonds zur Förderung der Wissenschaftlichen Forschung
(grant no. 200021-200517 / 1), and the Swiss Polar Institute (SPI) supports the development of the TEMPERA-C radiometer. We thank the
ARTS developer team for their support and Richard Larsson for implementing the Zeeman effect in ARTS. We also thank Francisco Navas-

Guzmán and MeteoSwiss for hosting the instrument and calibrating the noise diode, during the period when TEMPERA was located at the





MeteoSwiss technical centre in Payerne. Scientific color maps (Crameri et al., 2020) are used in this study to prevent visual distortion of the data and exclusion of readers with colour-vision deficiencies.



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
