# Peer review of "Thermal tides in the middle atmosphere at mid-latitudes measured with a ground-based microwave Radiometer"

_Atmospheric Measurement Techniques, 2024_

## Author Comment (AC1)

**Response to referee 1**

**1. GENERAL REMARKS**

- *The instrument apparently no longer does tipping curves. Are these not necessary to establish the level of tropospheric attenuation of the stratospheric measurement?*

The current focus of interest is stratospheric dynamic. Therefore, the latest operational mode performs without tipping curve, to maximise the observation time for stratospheric retrievals. Details on the tropospheric correction can be found below.

- *The diurnal temperature tides do not appear unreasonable, but, given that the measurements need to be taken from an instrument at a diurnally varying surface through a diurnally varying troposphere, are there any steps taken to ensure that these tropospheric variations are not mapped into the small (<1%) stratospheric variation shown in Figure 6? Perhaps the errors have been estimated and are much smaller than 1%, but, in any case, some short discussion of this would be appropriate.*

According to radiative transfer principals, the observed spectra can be separated into an tropospheric contribution ($0_2$ complex Figure. 2) and fine structure features ("Spikes" in Figure. 3), containing the stratospheric information. In the previous version of the retrieval the tropospheric contribution was corrected, using the line-wings according to (`https://doi.org/10.1029/98RS01000`). In the present version of the retrieval, the tropospheric profile is retrieved simultaneously with the stratospheric profile also by using the line wings. This is mentioned in Section 3 and we added further discussion in section 4. The accuracy of the troposheric profile is not enough for weather forecasting, but this method is much more sensitive for tropospheric variations than the previous version. Also, the current surface temperature is used in the apriori profile. However, during very cloudy or raining conditions the observations are removed from the dataset. In addition the ASF algorithm fits the spectral parameters over a 4-day sliding window, very short time scale variations are further reduced by this method. The tropospheric influence can also be estimated by looking at the rows of the AVK matrix, for stratospheric grid points but at tropospheric altitudes (Figure 1 in this document, not included in the manuscript). A row of the AVK matrix mirrors how much a specific grid point is influenced by grid points from other altitudes. Note the x-axis scale compared to Figure. 4 in the manuscript. Almost all values are below 1% up to around 9 km.

**2. SPECIFIC REMARKS**

- *Please present Figure 7 and Figure 8 in the same aspect ratio so that they can be more easily be compared.*

Figures were re-scaled

- *The result that stands out in these figures is the consistency of the 18 LST phase in the upper stratosphere and lower mesosphere. This recent article by Leroy and Gleisner seems to agree: https://doi.org/10.1029/2021EA002011*

Reference was included.

- *Line 241- "Mai" should be "May"*

Term was corrected

- *Paragraph beginning on line 248 and Figure 11- According to the text Figure 11 is a correlation between ozone mixing ratio and tidal amplitudes. If that is correct then please state it clearly in the caption as well. I don't understand the relevance of the ozone diurnal cycle mentioned in the first line of the paragraph. If ozone is driving a diurnal temperature variation I would think that this is related to the presence of solar irradiation during the day and has nothing to do with the small diurnal ozone variation. The final sentence of this paragraph is also confusing in this regard.*

  The correlation is made between ozone VMR and atmospheric temperatures, we added "VMR" in the caption.

  The forcing driven by the ozone diurnal cycle with a VMR variability of about 2% will be small. This part was meant as a motivation for further studies. We are currently, evaluating heating rates from WACCM and other data sources to estimate the energy fluxes. But a detailed analysis is beyond the scope of the paper. We have revised the main text.

- *"was located partially" should be "was*

  Term was corrected.

- *Line 110 "frequency stretch"? What does this mean?*

  The bandwidth of the spectrometer channels can have a small error ("frequency stretch") which is accounted for by including this quantity in the ensemble of retrieval quantities.

- *Figures 2 and 3 are referred to before Figure 1.*

  Figure was moved to position 3.

- *Figure 5 – Any comment on why the there is such a very low altitude peak in the earliest data (Feb. 2014?)*

  The time series starts in January 2014. The anomaly the reviwers refers to belongs to a FSW (Final Startospheric Warming) event and the corresponding planetary wave activity (https://egusphere.copernicus.org/preprints/2024/egusphere-2024-65/).

[Figure]

**Fig. 1**. Rows of the AVK matrix for stratospheric grid points, plotted at tropospheric altitudes the x-axis scale was reduced in comparison to Figure. 4 in the manuscript.

---

## Author Comment (AC2)

**Response to referee 2**

**1. GENERAL REMARKS**

- *The authors emphasize the advantage of local quasi-continuous temperature measurements against satellite data that typically need several weeks for full diurnal coverage. On the other hand, they only present monthly or seasonally averaged data. All variabilities are "hidden" in the error bars that include both instrumental errors as well as natural variabilities. I recommend showing at least one representative multi-day case that displays the need for the study of tides with high temporal resolution.*

The high temporal resolution is necessary for the spectral decomposition. Figure 6 shows multiple panels of a 14-day time series of 1 temperature a), temperature anomaly b), Temperature anomaly at fixed altitude and diurnal tidal amplitude obtained from the ASF2D d) with a temporal resolution of 3h for panels a) -c) and daily mean amplitudes for panel d).

There are different types of error bars shown in the manuscript: The retrieval provides information about the smoothing and measurement error (statistical uncertainties) for each profile and altitude. These errors are the basis for the error propagation in the adaptive spectral filter (ASF2D). The errors shown in Figure 7+8 are based on this error propagation. The geophysical variability is presented in Figure 9+10 as error-bar. This is now clarified in the text.

- *The authors claim in the abstract and the main text, an altitude coverage of 25 km to 55 km. a) I assume some kind of weather dependence. Is all data discarded that does not cover that range? b) All plots cover the range between 20 km and 60 km and features below 25 km are described as well without discussion (e.g., line 190). Please make clear which data can be used for tidal analysis. All other data should be omitted or clearly marked in the plots.*

There is a convention that a measurement response of 0.6 is considered to be required for a reliable and almost apriori independent retrieval. However, a measurement response below 0.6 does not imply that the data does not contain geophysical information (see Figure. 6 b) ). For illustration purposes, we plotted all the figures from 20-60 km. We adjusted the height range to 25-50 km everywhere in the main text. In the figures, we added bars at these altitudes to remind the reader about the effective height range.

**2. SPECIFIC REMARKS**

- *l. 55: I recommend providing the coordinates in degrees and decimals (as for the remaining text), but not in degrees and minutes.*

The units have been adjusted.

- *l. 114 and Figure 4: The MR>0.6 is only fulfilled below 49 km. This is in contradiction to the "used" data coverage of 25-55 km and to the display of data up to 60 km. Please clarify.*

We harmonized the altitude statements throughout the manuscript. The measurement response is a statistical quantifier of the retrieved solution compared to apriori information. Our apriori profile has no tidal information included and is constant during each day. Thus, tidal phase and amplitude are still valuable even in regions with a measurement response below 0.6 (Figure.6 b)).

- *l. 119-125: This paragraph partly repeats Section 2. I recommend shortening it and moving the reference to Figure 3 into Section 2.*

We will shorten this paragraph as suggested.

- *l. 126: Obviously, there is some weather dependence on the data coverage, even if much smaller than, e.g., with lidars. Please, provide some numbers for the typical data coverage (resolved by season or month). What is the minimal accepted data coverage per day?*

The data coverage is indeed dependent on weather conditions, namely cloud formation and rainfall affect the measurements. In the first step, affected spectra are filtered before an integration of 3h, if more than 10% of the spectra are filtered, the whole 3-hour spectrum is removed from the dataset. The procedure is chosen to produce an equal-spaced sampling time, which is important for spectral decomposition. A second filter filters out layer profiles after the retrieval. The amount of removed spectra is around 15% on average. The ASF algorithm handles data gaps and adjusts the error accordingly. We added further information in the main text.

- *Figure 8: If the phase plots belong to the 24-hour period, I recommend aligning them with the amplitude plots in the first row.*

Plots were aligned.

- *Fig. 7+8 and l. 175: I wonder whether the decrease in amplitude above 45 km is partly instrumental. Fig. 4 shows a FWHM of the AVK of > 15 km for most altitudes. Additionally, AVK and MR values decrease above 40 km, indicating an increasing contribution by the (tide-free) apriori. Please discuss.*

The amplitude profiles are most likely smoothed with the measurement response curve. We have included this in the main text.

- *Fig. 7+8 etc. and line 178: Line 178 suggests that the error bars show the standard deviations, i.e. they include natural variability and the (presumably much smaller) error from the ASF (line 153). Please mention explicitly what is shown. The natural variability should not be described as "error". On the other hand, the error of the mean would be a good indicator of the error of the seasonal amplitude or phase values. Furthermore, above 40 km the amplitude error and the phase error are quite high. This makes the amplitude decrease above 40/45 km as well as the constant phase questionable (if not instrumentally induced anyway). Please discuss.*

This was indeed misleading, Fig 7+8 shows the propagated tide error while Fig.9+10 shows variability. We have clarified this in the main text and captions. The high error by a constant phase also surprised us a little bit. This error (as mentioned in the main text) is a numerical propagation of the retrieval error, which does not mean, that the values are "off" by this amount. The phase and tide error increases with altitude because the total retrieval error increases with altitude (Fig 4). The constancy and agreement with other studies of the phase suggest, that even though the values have a high error, they are not random. A holographic analysis also suggest how variable tidal phases are. The phase errors should also seen in the context of a) the temperature tides are reflecting a total tide, which is given by the superposition of migrating and non-migrating modes, and b) the tidal interaction to changes of the mean wind with altitude or longitude/latitude (https://acp.copernicus.org/articles/20/11979/2020/acp-20-11979-2020.html and https://angeo.copernicus.org/articles/37/581/2019/angeo-37-581-2019.html).

- *l. 179: You should not describe features below 25 km if the usable range starts above.*

We have adjusted this sentence

- *Fig. 9: Again, the error of the mean may be a better indicator here to differentiate true peaks from variability.*

  We have discussed, which indicator is most suitable to show in this figure and come to the conclusion, that the natural variability is the best, to plot along with the time-resolved series. The tide error is calculated by propagating the total retrieval error, which consists of a combination of the measurement and apriori error (Eq. 10) and is shown in Fig. 7+8. Therefore the tide error depends on the instrument, the retrieval algorithm, data gaps, and only to a part on the short scale (4-day sliding window) noise dependent on the goodness of the sinusoidal fit. Therefore we have decided, that the natural variability may be a better indicator here to differentiate true peaks from variability. A comparison of the statistical uncertainty obtained from the ASF and the geophysical variability can be found in https://angeo.copernicus.org/articles/35/711/2017/.

- *l. 188: If the (phase) results are contaminated by long-period gravity waves, I would assume the same for the amplitudes. Please discuss. The mean amplitudes of 12-h and 8-h variations are about half of the diurnal amplitude. I wonder why this has such a strong effect on the accuracy of the phase calculation. Please explain in more detail.*

  The tidal amplitudes are of course also affected by other geophysical effects, very short-scale variations are filtered by the ASF algorithm (depending on the sliding window length). The phase calculation, however is numerically a little delicate by itself since it is calculated with a tangent function. Also, a factor of 0.5 in tide amplitude is also a factor of 0.5 in signal-to-noise ratio what leads to higher errors in the sinusoidal fits making also the determination of the exact location of the peak more difficult. Tidal phase can be highly variable and this varibality is captured by the ASF (https://angeo.copernicus.org/articles/37/581/2019/angeo-37-581-2019.html).

- *l. 204-208: MERRA-2 is not used in the main section of the manuscript. I suggest either removing these lines or incorporating Appendix A into the main text.*

  This line was removed as suggested.

- *l. 225-227: I am sorry, but I get confused about which part of the data is trustworthy. I assume that the limitations of data quality are reflected in the error bars. I do not see a difference between 2014/15 and 2016/17. On the other hand, a deviation from the climatological mean would be an interesting result.*

  A instrumental caused suppression of tide amplitude peaks would be within the tide amplitude error. The mentioned features are no significant maxima in 2014 where the variability is also not at maximum in summer, an amplitude and variability maximum in summer 2015, variability maximum in summer 2016, amplitude maximum in spring and autumn of the years 2016 and 2017.

- *l. 231-242: I am sorry, but I am still confused. You argue that only the year 2015 confirms the Lindzen theory. But a few lines above you mention that the data quality in this period is reduced. Then this data is not a good proof for the theory. I do not understand the further arguments.*

  We reword this sentence, which caused some ambiguity. There was no intention to use Lindzen's theory as justification.

  *a) Why is the large variability an indicator of the agreement with the Lindzen theory? I assume that linear theory does not include much variability.*

  Increased variability values indicate, that tides with high amplitudes were present in the ensemble, from which the mean value was taken.

   *b) What is the conclusion from the convective gravity waves? Do they perturb/obscure the tidal signal (technical effect) or do they influence the propagation conditions for the tidal waves (geophysical effect)?*

Gravity waves play an important role during the summer. Thick convective clouds disturb our measurements generating gaps in the time series and the launched gravity waves perturb the Stratosphere and, thus, affecting the tidal signal. Although the adaptive spectral filter mitigates the problem, the combination of large amplitude gravity waves and data gaps pose an additional challenge in the data analysis.

*My last point: I do not see enhanced variability in summer. Fig. 9 is dominated by year-to-year variability. "Amplitude errors" in Fig. 7 do not vary with season. The yellow line (JJA) is right behind the other ones. Please explain.*

The error bars in Figure. 7 are mainly instrumental and therefore do not vary with season. The error bars in Figure. 9 however show maxima in spring, summer, or autumn depending on the year.

- *l. 243: A small amplitude by itself does not explain the noise in the phases. The signal-noise ratio matters more, which is influenced by, e.g., natural variability or other waves with similar periods.*

The term "signal-to-noise ratio" is indeed more suitable in this case.

- *l. 246/247: Do you suggest that the upward propagating tides are filtered at 35 km and new ones are formed above? Please comment. How does this (constant) phase progression compare to other observations? How reliable is the constant phase, given the large phase error shown in Fig. 7?*

To our knowledge, a new generation of Tides should form at altitudes between 30-50 km due to the increased Ozone amount at these altitudes. At the stratosphere, a superposition of tides can be present. Tides generated by the absorption of solar radiation from the troposphere propagate into the stratosphere, but still have rather small amplitudes. Superimposed are the tides generated by the absorption of ozone. The absorption of the radiation from ozone forcing a in-situ generated tide. There is almost no vertical phase progression due to the fast photo chemistry, which causes a direct response and forms a new chemical balance under the solar radiation leading to clear day and night change, but only to a rather small additional daily variation (Schranz et al., 2019). However, these two lines are a pure description of the measurements without any further suggestions.

- *l. 250/251: I understand that a detailed comparison (correlation) between ozone variation and temperature tides will be outside the scope of this paper. However, please briefly sketch how the ozone variation (and not e.g. the absorption of solar radiation) affects the temperature.*

With a daily VMR variation of about 2% the influence by solar heating is very small, but worth to investigate. We are currently, evaluating heating rates from WACCM and other data sources to estimate the energy fluxes. We revised the main text in this part.

- *l. 274: Please check the height range. For readers focusing on the conclusion, it should be noted that the new capabilities are only available since 2022.*

Height range and main text have been adjusted.

- *l. 287: You should not bring up new aspects in the conclusions. I suggest adding some text in the discussion with the pros and cons of radiometer and lidar for observation of tides.*

Changed as suggested.

- *l. 305: I would interpret this as a slight change in phase by 1-2 h between 40 and 55 km, rather than a "constant phase".*

Sentence has been adjusted.